# AlphaFold2 modeling and molecular dynamics simulations of an intrinsically disordered protein

Hao-Bo Guo[1,2☯]*, Baxter Huntington[1,3☯], Alexander Perminov[1,3☯], Kenya Smith[4☯], Nicholas Hastings[4☯], Patrick Dennis[1], Nancy Kelley-Loughnane[1]*, Rajiv Berry[1☯]*

**1** Material and Manufacturing Directorate, Air Force Research Laboratory, WPAFB, Mason, OH, United States of America, **2** UES Inc., Dayton, OH, United States of America, **3** Miami University, Oxford, OH, United States of America, **4** United States Air Force Academy, Colorado Springs, CO, United States of America

☯ These authors contributed equally to this work.
* haobo.guo.ctr@us.af.mil (HBG); nancy.kelley-loughnane.1@us.af.mil (NKL); rajiv.berry@us.af.mil (RB)

## Abstract

We use AlphaFold2 (AF2) to model the monomer and dimer structures of an intrinsically disordered protein (IDP), *Nvjp-1*, assisted by molecular dynamics (MD) simulations. We observe relatively rigid dimeric structures of *Nvjp-1* when compared with the monomer structures. We suggest that protein conformations from multiple AF2 models and those from MD trajectories exhibit a coherent trend: the conformations of an IDP are deviated from each other and the conformations of a well-folded protein are consistent with each other. We use a residue-residue interaction network (RIN) derived from the contact map which show that the residue-residue interactions in *Nvjp-1* are mainly transient; however, those in a well-folded protein are mainly persistent. Despite the variation in 3D shapes, we show that the AF2 models of both disordered and ordered proteins exhibit highly consistent profiles of the pLDDT (predicted local distance difference test) scores. These results indicate a potential protocol to justify the IDPs based on multiple AF2 models and MD simulations.

## 1. Introduction

Intrinsically disordered proteins/regions (IDPs/IDPRs) are proteins or protein regions that lack well-folded three-dimensional (3D) structures [1–3]. These proteins are also referred to as "natively disordered/unstructured/unfolded proteins" in the literature [4–6]. The widespread presence of IDPs in the protein world and their importance to cell functions have been appreciated only starting from the dawn of this century [7]. Nonetheless, the vital roles IDPs play in cells are numerous, such as the formation of biomolecular condensates (also known as membrane-less organelles) [8], serving as protein-protein interaction hubs [9], in cellular signaling and regulation [10], as well as in the evolution of multicellular life form and cell type specifications [11].

Despite the importance of IDPs/IDPRs and the numerous studies in this field, the meaning of "disorder" is sometimes unclear. This is highlighted by the various descriptions, e.g.,

**Data Availability Statement:** All data generated or analyzed during this study are included in this article and its Supporting Information files. R codes for residue-residue interaction network and secondary structure element analyses and

visualizations, as well as the 1000 AF2 (V2.2.2) models of both Nvjp-1 and T7RdhA can be downloaded from the GitHub repository: https://github.com/haoboguo/Nvjp-1.

**Funding:** This work was supported by funding from the Office of the Under Secretary of Defense for Research and Engineering (OUSD R&E), Applied Research for Advancement of S&T Priorities (ARAP) Program. The funders had no role in study design, data collection and analysis, decision to publish, or preparation of the manuscript.

**Competing interests:** The authors declare no competing interests.

"dancing protein clouds" [12], "fuzzy proteins" [13], and other names summarized in [6, 12], used to describe IDPs/IDPRs. Many studies use, a binary classification based on disorder predictions, i.e., a residue $R_i$ is disordered if its disorder content (i.e., the probability of being disordered) $D_i > 0.5$, and a protein is an IDP if more than 50% of the residues are disordered. In reality, proteins dynamically engage in their functions, constituting a structure-function continuum [12, 14]. Therefore, a binary (or probability) classification may not be well-suited to determine whether or not a protein is an IDP.

The rationale for over 100 protein intrinsic disorder predictors [15] comes from the hypothesis that residual disorder content is primarily determined by the protein sequence, in line with Anfinsen's protein sequence-structure dogma [16]. Anfinsen's dogma is also the target of the CASP (Critical Assessment of techniques for protein Structure Prediction) biennial competition [17]. In 2020, CASP14 announced the grand breakthrough of AlphaFold2 (AF2), which achieved the goal of accurately predicting the protein structure simply from its primary sequence, with accuracy comparable to experimental structures including those by X-ray crystallography and cryoEM [18]. Inspired by CASP and its significant contributions to Alpha-Fold2 [18–20], in 2021, the Critical Assessment of protein Intrinsic Disorder prediction (CAID) biennial competition was established, for benchmarking the disorder predictors to improve the protein intrinsic disorder predictions [21, 22]. However, what is the exact definition of disorder and how it is justified experimentally or theoretically? The DisProt database [23]. for example, is regarded as the ground truth for IDPs (a subset from DisProt was used as the target for CAID-1 [21]). From the prediction part, however, different predictors may yield significant variations (Fig 1A). It is worth noting that AF2 structure predictions, particularly, assisted by the pLDDT (predicted local distance difference test) scores, may serve as important measures of the per-residue disorder in proteins [24].

The present AF2 structure database contains over 200 million protein models covering nearly all known protein sequences [25], including the human proteome [26]. The protein conformational space of this database is enormous [27] (which is in line with a recent parallel perspective [28]): some match well with known experimental models deposited in the Protein Data Bank (PDB [29]) (the "good"); some deviate significantly from the PDB models (the "bad"), possibly owing to different conditions such as the native state conditions cannot be satisfied in the AF2 modeling [30]; moreover, many models are not well folded at all (the "ugly"). The "ugly" models, not surprisingly, are often related to IDPs.

The present work focuses on one example of an "ugly" models: *Nereis virens* jaw protein-1, or *Nvjp-1* [31], which is the predominant protein found in the marine polychaete *N. virens* jaw. The jaw of *N. virens* is made up of 90% (w/w) proteins, yet its mechanical properties in terms of hardness and stiffness are comparable to human dentin [32]. Moreover, these properties were shown to be modulated by Zn-binding [31], rendering *Nvjp-1* a unique candidate for dynamic sclerotization [33]. In the present work, we show that *Nvjp-1* is an IDP: first, the AF2 model of *Nvjp-1* possesses low per-residue pLDDT scores; second, high PAEs (predicted aligned errors) of all residues indicate the absence of correlated movements or interactions among the residues. Moreover, different AF2 models as well as protein conformations taken from the MD simulations deviate largely from each other. We also modeled a theoretical *Nvjp-1* homodimer using AlphaFold-Multimer [34]. Similar to the monomer models, the structures of homodimer models are not consistent with each other; even the two monomers in a selected dimer exhibit large root-mean-square deviation (RMSD) from each other after alignment. This is confirmed by structure snapshots from MD trajectories of the *Nvjp-1* dimer. The results from AF2 (and AlphaFold-Multimer) modeling and MD simulations, therefore, suggest an implication for "disorder" that is associated with the structural heterogeneity observed in the modeling and simulations.

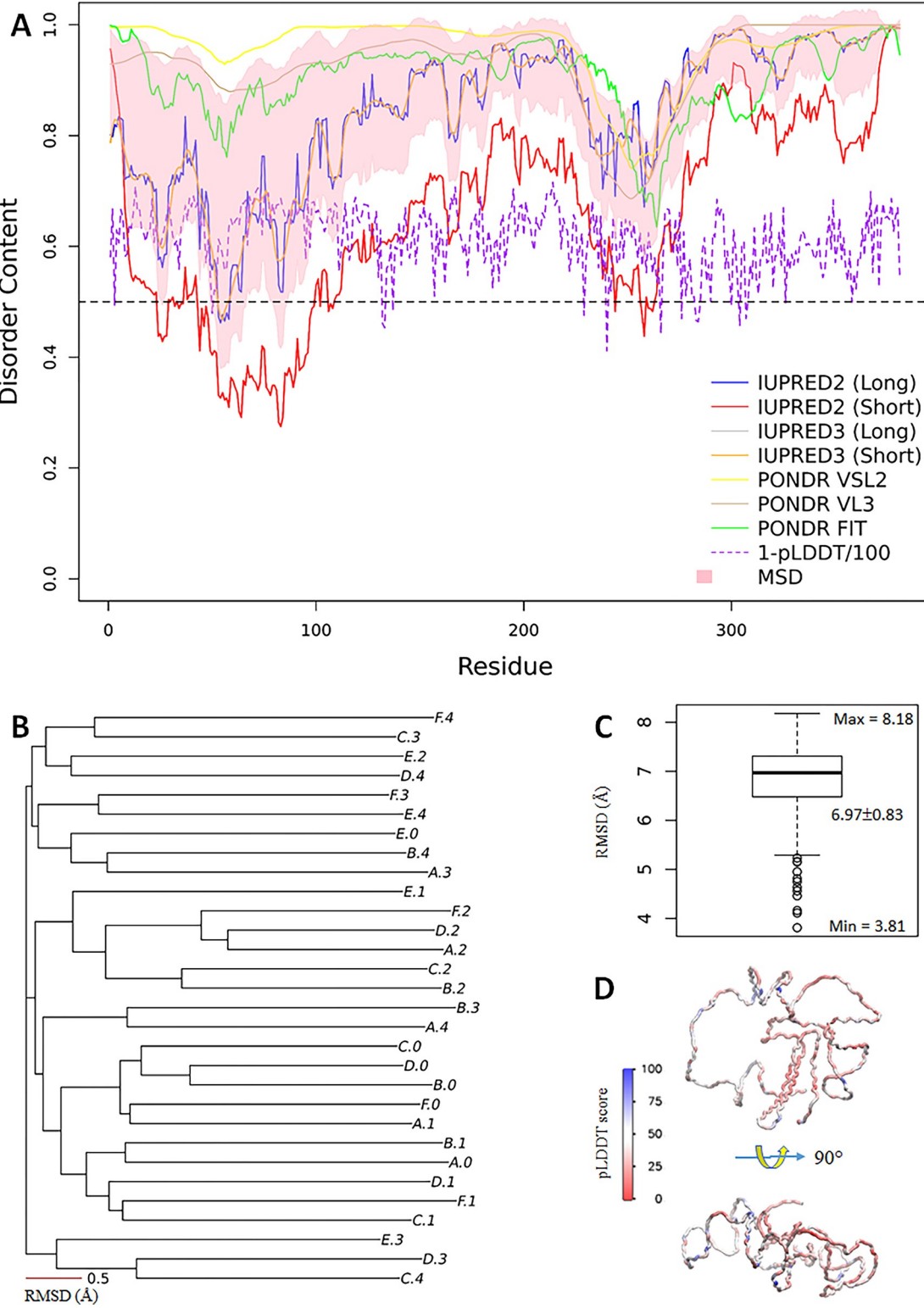

**Fig 1. *Nvjp-1* is an intrinsically disordered protein.** (**A**) Disorder contents ($D_i$) predicted by predictors IUPRED2A [51] (long/short), IUPRED3 [52] (long/short), PONDR VSL2 [53], PONDR VL3 [54], and PONDR FIT [55]. The mean±sd (MSD) from all 7 predictors are shown in pink shade. The purple dashed line is (1-pLDDT/100) using one of the models (A.0) for a comparison. The black dashed line at $D_i = 0.5$ is used for reference. (**B**) A phylogenetic tree reconstructed using the RMSD matrix calculated for 30 AF2 models. The scale bar (0.5 Å) serves as a reference for "atomic resolution" observed in a well-folded proteins which has the mean RMSD of 0.5 Å, see Fig 7. (**C**) A boxplot of the RMSD distributions. (**D**) Front (top) and side (bottom) views of one of the AF2 models colored by the pLDDT scores of all residues.

To represent ensembles of configurations of proteins by MD both sufficiently and correctly, long time-scale or enhanced sampling may be required [35] especially for addressing the ensemble-averaged experimental data [36]. A recent work indicated that a sufficient number of AF2 modeling can generate ensembles of protein conformations, containing "rare conformations" that might require considerably long time-scale MD to capture [37]. Here, besides the ensembles from independent MD simulations, 1000 AF2 models were further used to examine the structural heterogeneity of *Nvjp-1*, compared with 1000 AF2 models of a well-folded protein, the reductive dehalogenase *T7RdhA* [37]. The models of *T7RdhA* agree with each other (mean RMSD of 0.5 Å), whereas the *Nvjp-1* models deviate from other models (mean RMSD of 6.5 Å). Moreover, the results indicate that the pLDDT scores given by AF2 are highly consistent for both *Nvjp-1* and *T7RdhA*, which may therefore serve as a useful feature of the protein structures. We further constructed the residue-residue interaction networks (RINs) starting from the contact maps. The *Nvjp-1* RINs (1000 AF2 models) are dominated by transient interactions that are observed in less than 250 (<25%) AF2 models. In contrast, the *T7RdhA* RINs (1000 AF2 models) comprise mainly persistent interactions that are observed in more than 750 (>75%) AF2 models. Snapshots from MD trajectories exhibit the same patterns for the persistency of interactions in the RINs.

In the following, after summarizing the methods and simulation tools used (Section 2) we will show key results and discussion (Section 3) of this work. We show the intrinsic disorder profiles from different sequence-based predictors for *Nvjp-1*. The AF2 structure models of *Nvjp-1* significantly variate to each other; the *Nvjp-1* configurations from MD trajectories also differ significantly. The mean RMSD is ~7 Å of *Nvjp-1* between two AF2 models or two MD snapshots. Compared to a well-folded globular protein, *T7RdhA*, which has the mean RMSD of ~0.5 Å. Nevertheless, we show that the pLDDT profiles of both *Nvjp-1* and *T7RdhA* are highly consistent, indicating it may serve as a useful feature of proteins. We show a theoretical models of a doughnut-shaped *Nvjp-1* homodimer. We also show the distinct RIN patterns between the IDP (*Nvjp-1*) and the globular protein (*T7RdhA*). More analyses that support the conclusions of the present work (summarized in Section 4) can be found in the supporting information (SI). Potential future directions are also proposed in Section 4.

## 2. Methods and materials

### 2.1. Structure predictions and comparisons

The sequence of *Nvjp-1* protein (381 AA) is shown in the SI (S1 Fig). AlphaFold2 (V2.1) [18], together with AlphaFold-Multimer [34], are used for structure predictions of both the *Nvjp-1* monomer and dimer from the primary sequence. We constructed 1000 monomer and 25 dimer models in total. TM-align [38] and MM-align [39] are used to align the monomers and dimers, respectively, to calculate the root-mean-square deviations (RMSD) between each pair of proteins (single chain monomer or double chain dimer). We converted the RMSD matrices into tree-like representations (or clustering) using the R package *APE* [40], which utilize a neighbor-joining algorithm to inform trees from distance matrices in phylogenetics.

### 2.2. Molecular dynamics simulations

Molecular dynamics (MD) simulations are performed using the NAMD software [41, 42]. The CHARMM force field c36m [43]—which has been built to reflect residual flexibility in IDPs—is utilized for the protein, together with a modified TIP3P model for the solvent [44]. The water box for solvating is at least 15 Å larger than the protein in each of the six directions (X+, X−, Y+, Y−, Z+, and Z−). $Zn^{2+}$ and $Cl^-$ are used to neutralize the system at a concentration of 0.1 M. The Solvate and Autoionization packages of VMD software [45] are used for solvation and ionization, respectively. In the MD simulations, energy minimization was first performed

for 50,000 steps at 0 K. Then the system temperature is increased to 300 K at a rate of 0.001 K/step. A constant-pressure (1 atm), constant-temperature (300 K) NPT ensemble is used in the MD simulation maintained by the Langevin piston controls. The SHAKE algorithm is applied to constrain bonds with H atoms, and a 2 fs step length is used in the MD simulations. The non-bonded interaction cutoff switching is set at between 9 and 11 Å. The particle mesh Ewald summation with a grid spacing of 1.35 Å is applied for the long-range interactions. For each system, a 100 ns production run is performed after a 10 ns equilibration run. Approximately, the sidechain of a His residue has a $pK_a$ of ~6 [46], hence at neutral or basic pH (e.g., >7), either $N_{\varepsilon 2}$ or $N_{\delta 1}$ is protonated (neutral charge); whereas at acidic pH (e.g., <5), both $N_{\varepsilon 2}$ and $N_{\delta 1}$ are protonated (charge +1). As an IDP, all residues in *Nvjp-1* tend to be exposed in the bulk water owing to their high fluctuations. Most of the simulations were conducted under a neutral pH (~7), under which we used the state where only $N_{\varepsilon 2}$ is protonated but $N_{\delta 1}$ is not, which is referred to as **HSE**; we also performed MD of the *Nvjp-1* monomer under an acidic pH, under which both $N_{\varepsilon 2}$ and $N_{\delta 1}$ of all His residues are protonated and this state is referred to as **HSP**. Nine monomer trajectories (8 HSE and 1 HSP) and three dimer trajectories (all in HSE), 100 ns each, have been analyzed. We extended the MD simulation for each system to longer time scales, (680 ns for HSE, 800 ns for HSP and 1250 ns for the dimer, see Table 1), and analysis of the final 500 ns trajectories were used for analysis.

**Table 1. A summary of interaction numbers from MD simulations (9 for monomer and 3 for dimer).** The models shown in the main text are highlighted in bold font.

| Name[1] | State[2] | Atom#[3] | Length[4] | $R_{gyr}$[5] | State[6] | Total[7] | Persistant[7] | Transient[7] | Med.[7] |
|---|---|---|---|---|---|---|---|---|---|
| **Mono.A.0** | **Mono** | **280219** | **680 ns** | **28.9±1.5** | **HSE** | **4853** | **726** | **3376** | **751** |
| **Mono.A.0** | **Mono** | **277324** | **800 ns** | **41.3±0.9** | **HSP** | **5936** | **570** | **4818** | **548** |
| Mono.C.2 | Mono | 391696 | 100 ns | 34.1±1.1 | HSE | 4003 | 480 | 3048 | 475 |
| Mono.C.3 | Mono | 220786 | 100 ns | 28.1±0.6 | HSE | 3800 | 572 | 2573 | 655 |
| Mono.C.4 | Mono | 221281 | 100 ns | 26.6±0.4 | HSE | 3787 | 587 | 2554 | 646 |
| Mono.D.0 | Mono | 381121 | 100 ns | 31.2±1.0 | HSE | 3939 | 507 | 2829 | 603 |
| Mono.D.2 | Mono | 388780 | 100 ns | 41.5±0.7 | HSE | 3378 | 497 | 2309 | 572 |
| Mono.D.3 | Mono | 226690 | 100 ns | 26.6±0.8 | HSE | 3767 | 587 | 2498 | 682 |
| Mono.D.4 | Mono | 216493 | 100 ns | 25.1±0.6 | HSE | 4151 | 532 | 2975 | 644 |
| **Di.A.0** | **Dimer** | **123111** | **1250 ns** | **33.8±0.4** | **HSE** | **6847** | **2168** | **2838** | **1841** |
| Di.C.2 | Dimer | 136282 | 100 ns | 31.7±0.2 | HSE | 6154 | 1662 | 3131 | 1361 |
| Di.D.0 | Dimer | 147683 | 100 ns | 34.7±0.4 | HSE | 6380 | 1686 | 3334 | 1360 |
| AF2[8] | Mono | - | - | 49.2±12.4 | - | 5982 | 397 | 5388 | 197 |
| T7RdhA[9] | Mono | - | - | 20.9±0.04 | - | 2352 | 1850 | 411 | 91 |

1. The names (A.0 etc.) refer to the models shown in Figs 1 and 4 in the main text. Mono for monomers and Di for dimers, respectively.

2. Only monomers and dimers are modeled in this work, despite higher order of oligomers may exist.

3. The Atom number refers to the total number of atoms of the MD system, including water molecules and ions. It is notable that the monomer systems are significantly larger than the dimer systems, which is owing to the relatively large radius of gyration of the monomers, compared to the dimers.

4. Length of the MD simulation

5. The median±IQR of the radius of gyration (in Å) of the proteins (monomer or dimer) from the last 20 ns of the 100 ns MD trajectories, or the last 500 ns of the three long time-scale MD trajectories (in bold fonts).

6. As illustrated in the main text that a binary states of the His residues have been considered: under high pH, all His are mono-protonated on $N_{\varepsilon 2}$ (HSE state, neutral), whereas under low pH, all His are double-protonated at both $N_{\varepsilon 2}$ and $N_{\delta 1}$ (HSP state, charge +1).

7. Total edge numbers include all residue-residue interactions appear in MD trajectory; persistent edges (Strong) are those appear in >75% of all configurations of the MD trajectory (100 ns, or the last 500 ns for longer time-scale MD trajectories); transient edges are those only appear in <25% of all configurations of the MD trajectory; the medium (Med.) strength edges are all other interactions other than the strong and transient ones. The persistent, transient and medium strength edges are colored by red, blue and gray in the RIN figures.

8. From 1000 *Nvjp-1* AF2 (V2.2.2) models.

9. For a reference, data from 1000 AF2 (V2.2.2) models of a well-folded protein, T7RdhA [37].

## 2.3. Residue interaction network

Residue interaction networks (RINs) are constructed based on the contact maps to capture the interactions contributed from different residues. The RINs are constructed for each AF2 models, or for the MD trajectories, RINs are calculated from the snapshots taken every 1 ns after equilibrations. Details of the RIN construction are summarized in a previous work [37]. Briefly, the distance between the two residues $R_i$ and $R_j$ (denoted as $D_{ij}$) is defined as the shortest distance between all non-hydrogen atoms the two residues. For each AF2 model or a configuration taken from MD trajectories, an adjacency matrix is estimated from the distance matrix $[D_{ij}]$ under the cutoff of 3.5 Å: $A_{ij} = 1$ if $D_{ij}$ is less than 3.5 Å or $A_{ij} = 0$ otherwise. Without considering the directions (which may be important for certain interactions such as hydrogen-bonds) or weights (including attraction versus repulsion), this binary adjacency matrix can only be transformed into an undirected, unweighted network, or a contact map, in which the indices are amino acid residues and the edges are 1 for an interaction or 0 for no interaction. However, using ensembles of configurations (multiple AF2 models or snapshots from MD simulations), a weighted residue interaction network can be constructed by adding all binary adjacency matrices together. The *igraph* R package is used for network analysis [47]. The residue-residue distances are calculated using the *bio3D* R package [48]. An interaction is considered "persistent" if it appears in more than 75% of all models (either AF2 models or configurations from MD trajectories), and it is considered "transient" if it appears in less than 25% of all models. See Results and Table 1 for details.

## 2.4. Comparisons between Nvjp-1 and a well-folded protein T7RdhA

Besides the 30 *Nvjp-1* models using the original AF2 code (V2.0.1, summarized in Table 1 and Fig 1) [18], we further built 1000 AF2 models using a more recent AF2 code (V2.2.2) [18] and compared with 1000 AF2 models of a well-folded protein, the reductive dehalogenase from *TMED77 acidimicrobiaceae* (*T7RdhA*) [37].

## 2.5. Other tools

The STRIDE program is used to predict the secondary structure elements in the proteins via the *bio3d* R package [49]. RGN2, a recent deep-learning tool using a language model is also applied to predict the *Nvjp-1* structure [50].

## 3. Results and discussion

### 3.1. Nvjp-1 is an intrinsically disordered protein

We used different predictors to evaluate the intrinsic disorder contents ($D_i$) of *Nvjp-1* (Fig 1A). The disorder profiles are consistent: the Pearson's correlation coefficients, PCC = 0.44±0.51 (median±IQR), and median p-value of $2.0 \times 10^{-19}$, with nearly all residues having a $D_i > 0.5$; the mean disorder profile from all 7 predictors has a $D_i = 0.85 \pm 0.09$, indicating that *Nvjp-1* protein is a fully disordered protein. We used AF2 to model the *Nvjp-1* structures: 6 independent runs (A to F) produced 5 models (0 to 4) each, yielding 30 AF2 models in total. We noticed that each of the 30 models structurally deviated from the other models. To verify this, we measured the RMSDs among all 30 protein models using TM-align, and converted the RMSD matrix (30×30) to a phylogenetic tree (Fig 1B). The RMSD matrix has values of 6.97±0.83 Å, confirming the above observation. The structure of one of the models (A.0) is shown in Fig 1D, colored by the per-residue pLDDT scores, which indicates the low confidences given by AF2. Moreover, the pLDDT scores provided by AF2 do not correlate with any of the disorder profiles

assessed here. As a comparison, in a recent work [37], 320 AF2 models of a well-folded globular protein have overall RMSDs of 0.47±0.15 Å, or at the "atomic accuracy" [19].

## 3.2. Structural heterogeneity of Nvjp-1 monomer and the pH effect

For a globular protein, multiple AF2 models agree with the configuration space sampled by MD simulations [37], indicating the ability of AF2 for capturing different, and (potentially) biologically relevant, conformations. However, for *Nvjp-1*, it is difficult to tell which model generated by AF2 or which snapshot from MD simulation is of relevance. For the monomers, starting from a same AF2 structure (A.0 in Fig 1B), we perform two independent simulations with a) all His residues double protonated (i.e., in +1 charge states) or single protonated (neutral charge), to mimic acidic (pH 3) and basic (pH 8) conditions, respectively. It has been shown that *Nvjp-1* dissolves at low pH but precipitates at high pH, mainly owing to the high percentage (27 mol%) of histidine residues [33]. A recent report using MD simulation indicates that *Nvjp-1* has a larger radius of gyration ($R_{gyr}$) at low pH (3) compared to high pH (8) [56]. Consistent with this report [56], our simulations (taken from the final 500 ns from long MD trajectories) show that *Nvjp-1* monomer has a considerably larger $R_{gyr}$ (41.3±0.9 Å) at the HSP state (acidic pH) compared to HSE state (neutral pH, $R_{gyr}$ = 28.9±1.5 Å), likely caused by the repulsions among the positively charged His residues (Fig 2A). Note that the Asp and Glu residues remain deprotonated in both conditions in this work, despite their $pK_a$ values. However, the initial AF2 models (before MD simulations) have even larger $R_{gyr}$ (49.2±11.0 Å from 30 models generated by AF2 2.0.1, or 49.2 ±12.4 Å from 1000 models generated by AF2 V2.2.2) values. The $R_{gyr}$ of the initial AF2 and MD simulation models are larger than the $R_{gyr}$ computed in the previous *Nvjp-1* monomer model (~22 Å) which utilized a polarizable force field to conduct the MD simulations [56]. For a comparison, the $R_{gyr}$ of a well-folded protein, T7RdhA [37], is 20.9±0.04 Å (Table 1).

We further compared the configurations taken from MD trajectories at low or high pH). The configurations significantly deviate from other configurations collected from the same MD trajectory (Fig 2B), and all pairs of structures show large RMSDs (7.14±0.57 Å, Fig 2C). Similar to Fig 1, the scale bar in Fig 2B (0.5 Å) serves as a reference for the "atomic resolution". The snapshots after 100 ns MD simulations of both high (top) and low (bottom) pH MD are shown in Fig 2D. Note that in contrast to previous studies [32, 56], little or no apparent secondary structures appear in *Nvjp-1* monomer during the MD simulations, supporting the fact that *Nvjp-1* is an IDP.

## 3.3. PAE map, distance map and residue interaction network

For the constructed models, AF2 provides the predicted aligned error (PAE) map to quantify the potential distance errors in the model: [18] if the i-th residue is aligned to the "real" (ground truth) model, the (potential) error, or the PAE value at the j-th residue is $P_{ij}$ (in Å). The PAE matrix is asymmetric, i.e., $P_{ij}$ and $P_{ji}$ may not be equal. We previously suggested that the PAE map originates from the protein dynamics [57]. Here, we show that the PAE map is highly consistent with the distance map calculated from the MD trajectory (under both high and low pH, Fig 3A–3C). The distance maps can be converted to a binary contact map, then further visualized as residue interaction networks (RINs, see Methods section). In this network, the vertices are the residues and the edges are the interactions: an interaction exists if the shortest distance between the non-H atoms of residues $R_i$ and $R_j$ both residues is shorter than 3.5 Å [37]. The RINs indicate that in addition to the backbone-interactions (e.g., two neighboring residues are connected during the MD simulation), the majority of residue-residue interactions are short-lived, or transient, which are colored in blue in Fig 3D/3E, or S2 Fig in the SI. It

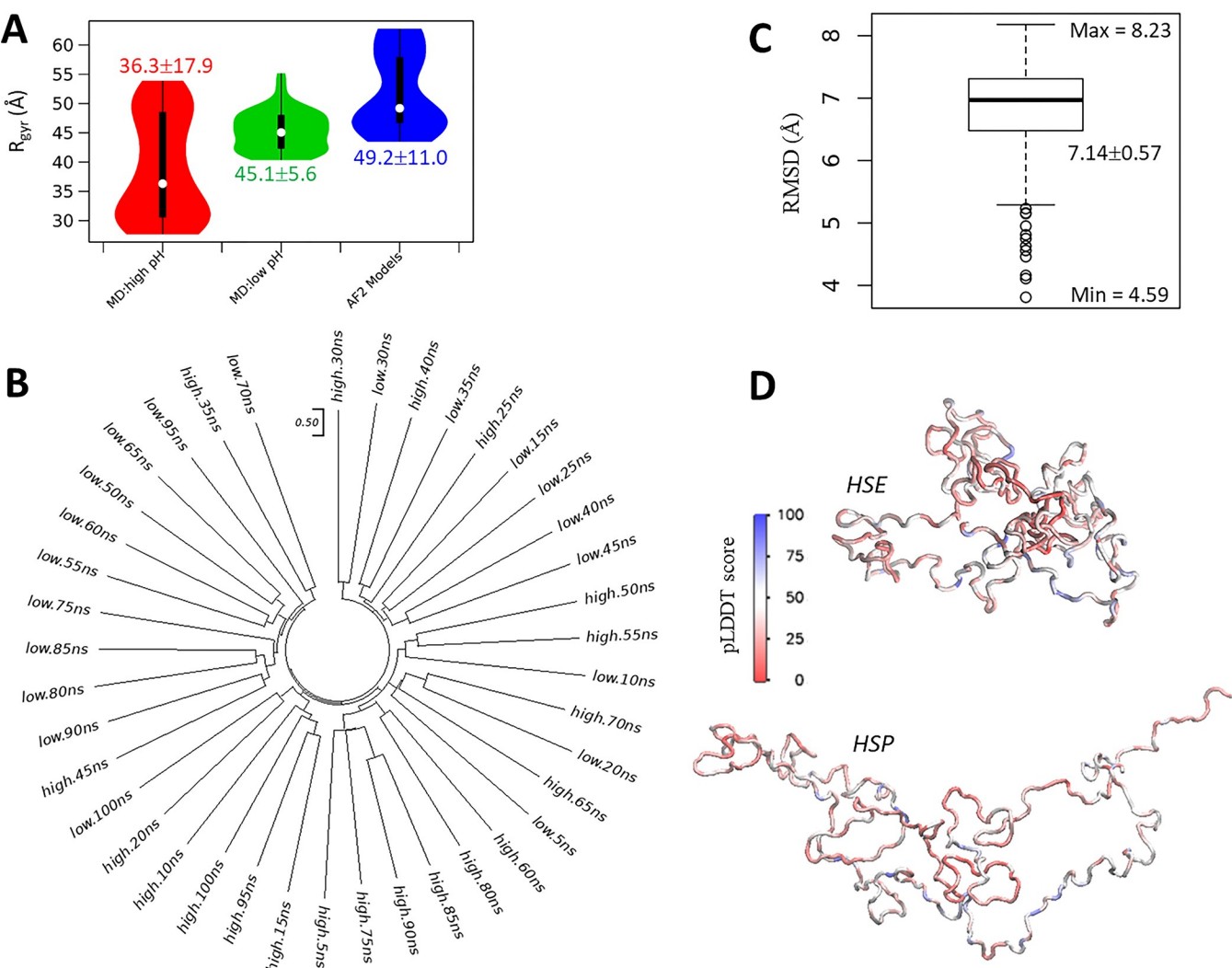

**Fig 2. The effect of pH on Nvjp-1 monomer structure.** (**A**) Violin plots for distribution of radius of gyration ($R_{rgy}$, Å) of protein structures taken from MD simulations under high pH (red, left), low pH (green, middle), compared with the AF2 models shown in Fig 1. (**B**) Structure-based phylogenetic tree using snapshots taken from 100 ns MD simulations under both HSE (neutral pH) and HSP (acidic pH) states. A scale-bar of RMSD = 0.50 Å is provided as a reference for the "atomic resolution" observed in well-folded proteins. (**C**) A boxplot of the RMSD values calculated from all pairs of structures in B. (**D**) *Nvjp-1* monomer snapshots after 100 ns MD performed under HSE (neutral pH, top) and HSP (acidic pH, bottom) states. The structures are colored by the pLDDT scores of the original AF2 structure (A.0 in Fig 1) before the MD simulation.

is interesting that the residue-residue interactions are dominated by the contributions from both Gly and His, the two most abundant residues in *Nvjp-1*. In addition, under HSE state (neutral pH), the $R_{gyr}$ of *Nvjp-1* monomer becomes smaller, and more residue interactions are observed compared to HSP state, or acidic pH conditions.

The results shown above are based on MD simulations from the A.0 structure in Fig 1B. Despite the structural heterogeneity, the results based on other structures are largely consistent, see S3 Fig in the SI for the results from two other models.

### 3.4. Nvjp-1 dimer: AF2 structure, distance map and RIN

In protein purification, previous experimental work (Dennis et al., unpublished) observed a potential dimer band from SDS-PAGE, which can sustain high temperature in SDS detergent

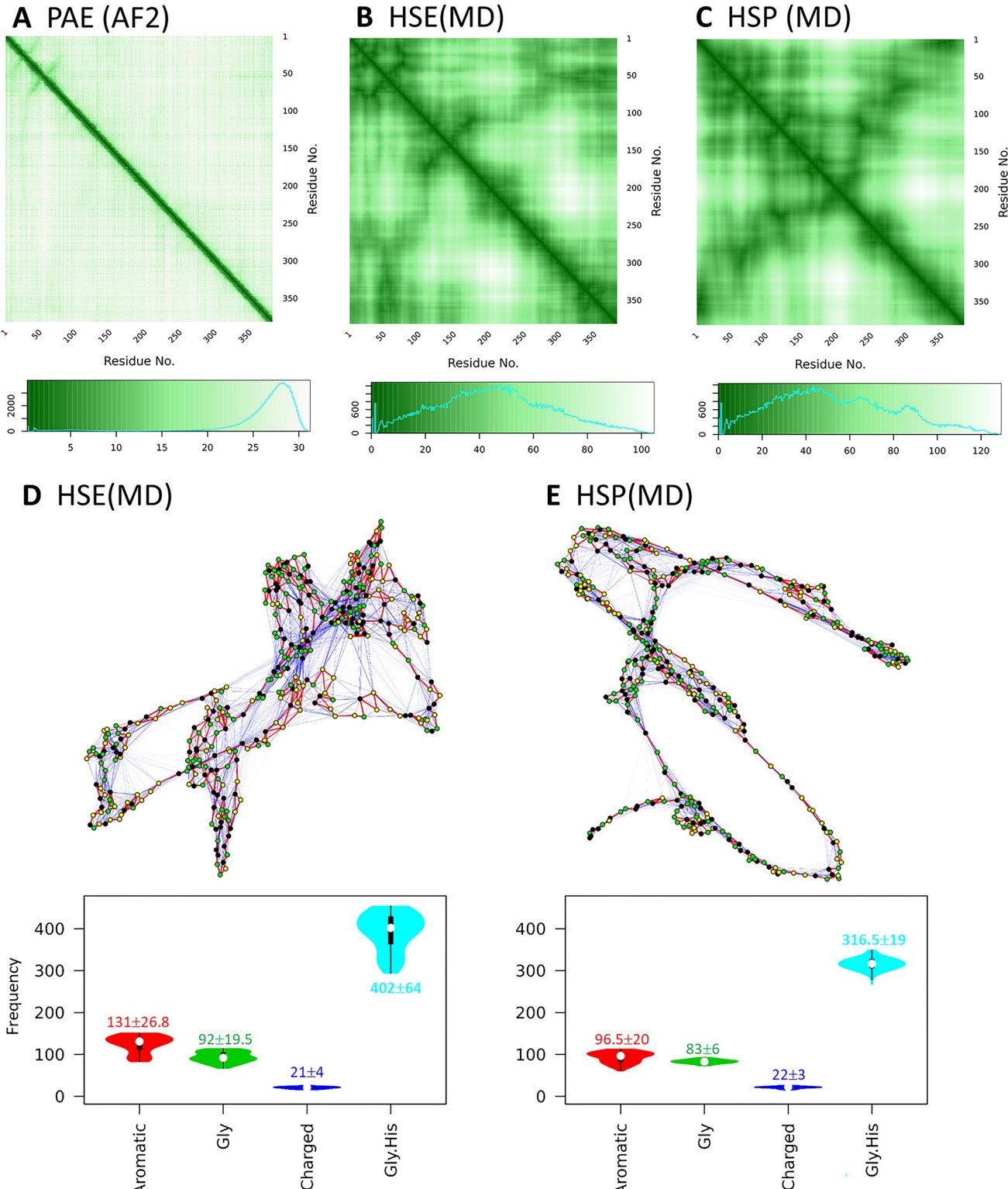

**Fig 3. PAE, distance map and residue interaction network of *Nvjp-1* monomer.** (**A**) PAE (predicted aligned error) map of one of the AF2 models (A.0 in Fig 1). The distance maps averaged from 100 ns MD simulations under (**B**) HSE (neutral pH) state and (**C**) HSP (acidic pH) state. Weighted residue interaction networks (RINs) under (**D**) HSE and (**E**) HSP states. In the RINs, green nodes are Gly, black nodes are aromatic (H, Y and F) residues of *Nvjp-1*. In the MD trajectory, the edges (3821 for HSE and 3295 for HSP) appear in >75% of all configurations are persistent interactions, and are shown in red (518, 13.6% for HSE; and 488, 14.8% for HSP), edges appear in <25% of all configurations are transient interactions and are in blue (2733, 71.5% for HSE;

and 2320, 70.4% for HSP), and those in between are in gray (570, 14.9% for HSE; and 487, 14.8% for HSP). A circular representation of the RINs can be found in S2 Fig of the SI. Violin plots of interaction frequencies contributed from aromatic residues (H, Y, F), Glycine, charged residues (D, E, R, K) or Glycine and Histidine are shown. Median±IQR are shown for the interactions during the 100 ns MD.

(unpublished). Here, using AF2-multimer, theoretical *Nvjp-1* homodimer models were predicted by AlphaFold-Multimer, which are doughnut-shaped (Fig 4). Similar to the monomer, the dimeric structures (25 models in total) deviated from each other with an RMSD of 6.91 ±0.65 Å (Fig 4A/4B). In addition, little or no apparent secondary structural elements were observed in the *Nvjp-1* dimers (Fig 4C). We compared the structural similarities among individual monomers constituting the dimers (50 monomers in total) and observed the same trend for the single *Nvjp-1* chains within the dimers, i.e., all individual dimer chains showed distinct structures, with an RMSD of 6.91±0.72 Å (S4 Fig in SI).

The AlphaFold-Multimer models of *Nvjp-1* homodimers have a smaller $R_{gyr}$ compared to the AF2 models of *Nvjp-1* monomers. Therefore, the *Nvjp-1* monomers are considerably larger than the dimers (Table 1). Interestingly, during the MD simulations, the $R_{gyr}$ of the *Nvjp-1* homodimer is larger than the monomer at high pH, but is smaller than the monomer at low

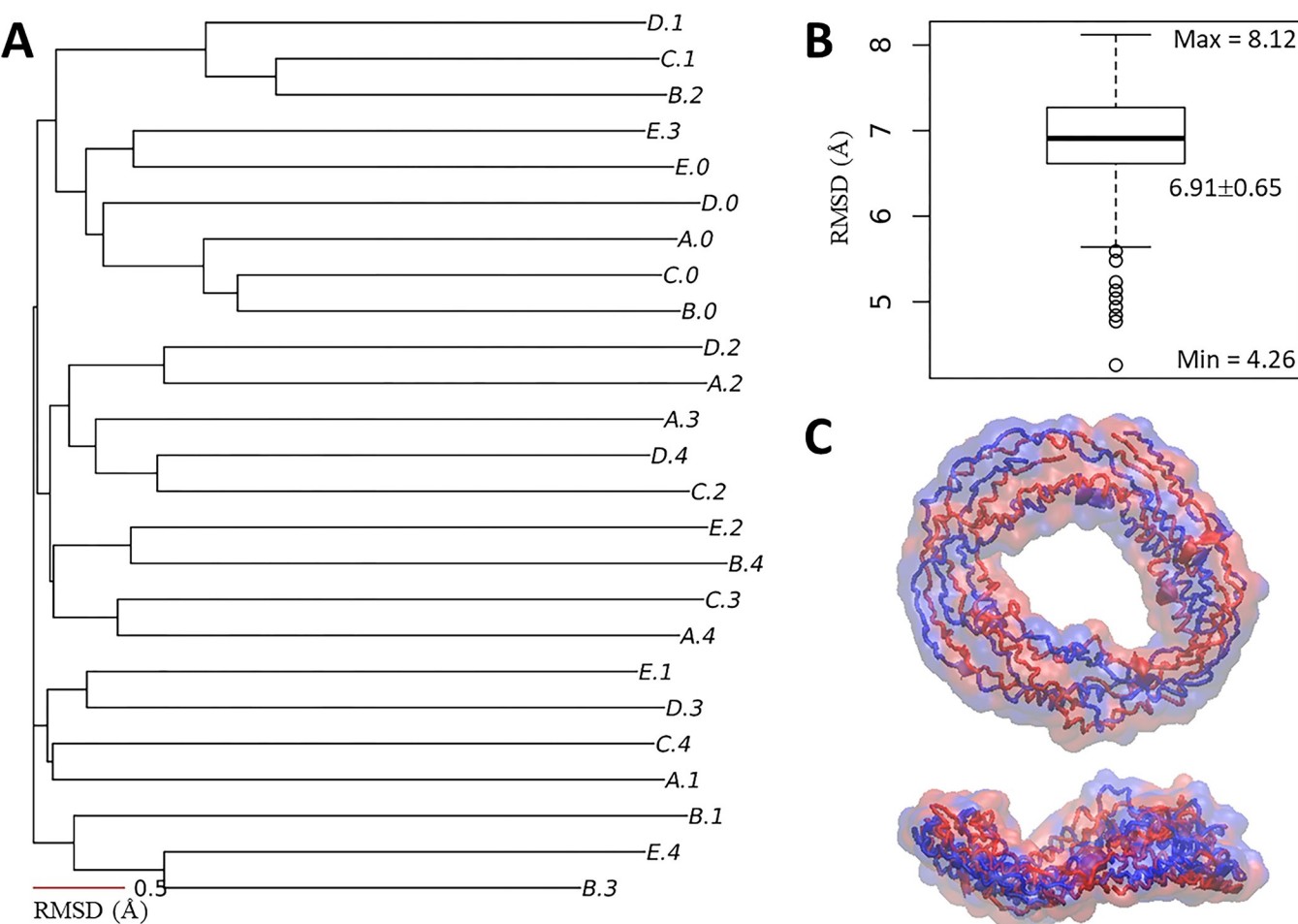

**Fig 4. *Nvjp-1* homodimer structure.** (**A**) Structure-based phylogenetic tree of 25 AF2 models from five independent runs. (**B**) A box plot of the RMSD matrix for the tree in A. (**C**) The doughnut-shaped *Nvjp-1* dimer structure; top: front view, bottom: side view. The two chains of the dimer are colored in blue and red, respectively.

pH. However, the fluctuation of $R_{gyr}$ of the *Nvjp-1* dimer is considerably smaller than that of the monomer under both pH conditions (Fig 5A), indicating that the *Nvjp-1* dimer is relatively rigid compared to the monomers. The $R_{gyr}$ of the monomers and dimers of MD trajectories of

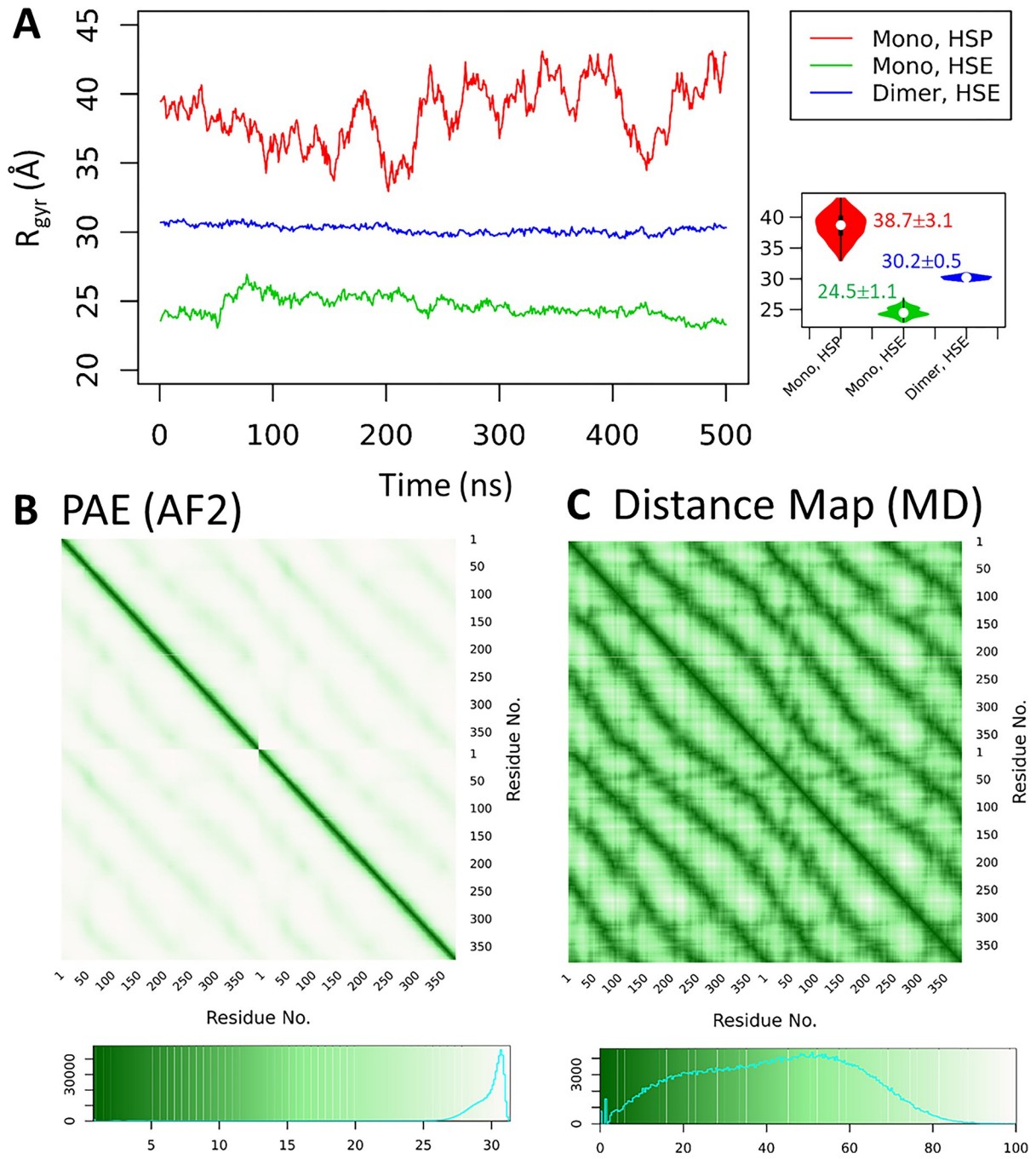

**Fig 5. A rigid *Nvjp-1* dimer.** (**A**) The radius of gyration ($R_{gyr}$) of the dimer is larger than the monomer under HSE state but smaller than the monomer under HSP state. Smaller fluctuations of $R_{gyr}$ in the dimer indicates this structure is relatively rigid compared to the monomers (under both HSE and HSP states). The inset shows violin plots of the $R_{gyr}$. (**B**) The PAE map from AF2 and (**C**) the distance map averaged from the snapshots from a 500 ns MD exhibit a consistent pattern indicating each of the two chains loops for two circles in the doughnut-shaped *Nvjp-1* dimer.

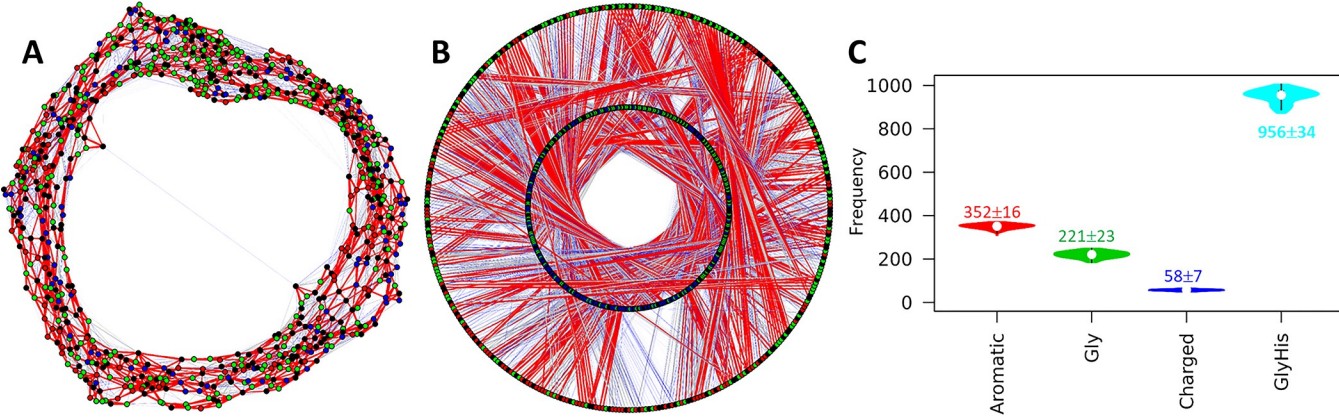

**Fig 6. Residue interaction network (RIN) of *Nvjp-1* dimer.** (**A**) A randomized RIN and (**B**) a circular representation of the RIN of *Nvjp-1* dimer. The Gly and His residues are in green and black, and other residues from chain A and B are in blue and red, respectively. In the circular presentation, nodes in the inner circle are from chain A and nodes in the outer circle are from chain B, respectively. The RIN has 6847 edges in total. In the MD trajectory, the edges that appear in >75% of configurations (strong interactions) are shown in red (2168, 31.7%), the edges that appear in <25% of all configurations (transient interactions) are in blue (2838, 41.4%), and those in between are in gray (1841, 26.9%). (**C**) Violin plots of interaction frequencies contributed from aromatic residues (H, Y, F), Glycine, charged residues (D, E, R, K) or Glycine and Histidine are shown. Median±IQR are shown for the interactions during the 100 ns MD. The RINs of two other *Nvjp-1* dimer models (Table 1) are shown in S5 Fig of SI.

other systems are summarized in Table 1. The inconsistency of MD simulations may also be owing to the intrinsic disorder nature of the *Nvjp-1* protein, however, for the dimers, the relatively small fluctuations in $R_{gyr}$ suggest a stabilization effect upon dimerization. The last 500 ns of the MD trajectories (680–1250 ns) were used for Fig 5A.

Further comparison of the *Nvjp-1* dimer PAE and distance maps demonstrated an apparent pattern (Fig 5B/5C). In addition to a strong diagonal line contributed by the interactions (or affiliations) of consecutive residues, which is also seen in the PAE and distance maps of the monomers (Fig 3A–3C), a periodic interaction trend emerges from each chain as it loops twice around the dimer doughnut (Fig 5B and 5C).

We constructed the RIN of the *Nvjp-1* dimer using the same criterion as the monomer (Fig 6) and observed the total interaction number of the dimer RIN was roughly double that of the monomer RIN (6847 for dimer model A.0, and 3821 for monomer model A.0 at HSE state). However, the interactions that persisted in over 50% of the configurations (i.e., strong interactions), taken from the MD trajectory, significantly increased in the dimer. After removing the backbone interactions (i.e., interactions from consecutive residues), the *Nvjp-1* dimer demonstrated 1408 strong interactions, which was over four times than those in the monomer (343 at high pH and 293 at HSP state). Moreover, the monomer had a higher ratio (71.5% at HSE state, 70.4% at HSP state) of transient interactions than that of the dimer (41.4%, HSE). See Table 1 for a summary of interaction numbers from the MD simulations. The RINs of two other *Nvjp-1* dimer models are shown in S5 Fig of SI.

### 3.5. Consistent pLDDT profiles predicted by AF2

Above discussions showed the inconsistency in the AF2 structure predictions of *Nvjp-1*, and possibly of other IDPs. In contrast, we recently found that multiple AF2 models of a well-folded protein, *T7RdhA*, are highly consistent, with mean RMSD lower than 1 Å, i.e., within "atomic resolution" [19]. We reconstructed 1000 AF2 models for both the disordered (*Nvjp-1*) and ordered (*T7RdhA*) proteins, and plotted the RMSD profiles against a randomly chosen model (Fig 7A). The large RMSD values of *Nvjp-1* models (6.5±1.1 Å, also see Fig 2), compared to those of *T7RdhA* models (0.5±0.1 Å) suggest that an ensemble of structures of an IDP (such

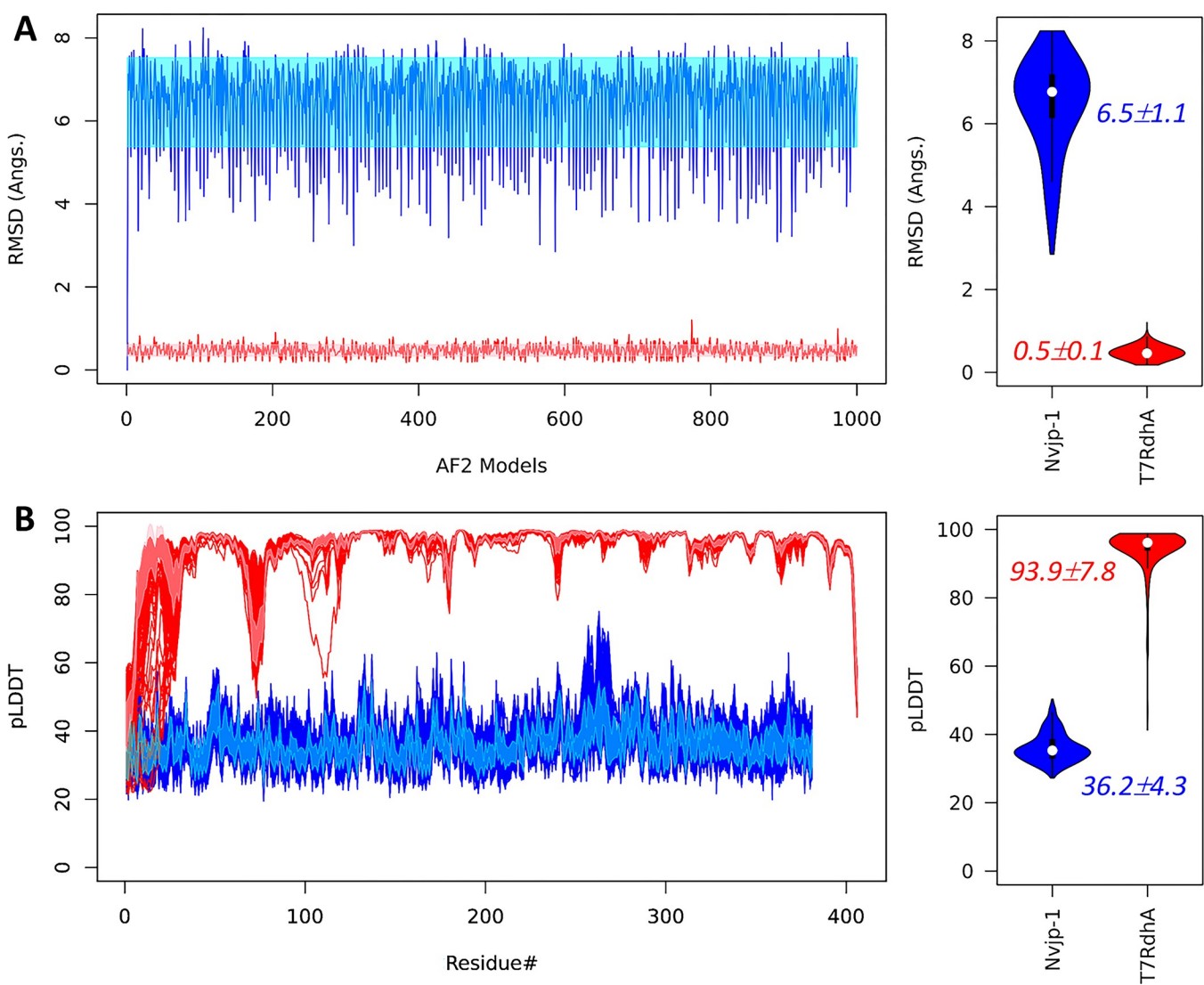

**Fig 7.** (**A**) RMSD fluctuations of 1000 AF2 models against a randomly selected model indicate high variation in structures of *Nvjp-1* (blue), and high consistency in structures of a well-folded protein, *T7RdhA* (red); (**B**) the per-residue pLDDT scores of *Nvjp-1* (381 AA, blue) and *T7RdhA* (406 AA, red) of 1000 AF2 models. The shaded areas of *Nvjp-1* (cyan) and *T7RdhA* (pink) profiles in the left panels illustrate the mean±sd. Violin plots of the RMSD and mean pLDDT profiles are shown in the right panels for both *Nvjp-1* (blue) and *T7RdhA* (red).

as *Nvjp-1*), without inferences from other molecules, may sample a relatively diverse conformational space, either from AF2 structure modeling (Figs 1, 4 and 7) or from MD simulations (Fig 2). Whereas an ensemble of structures of an ordered protein (such as *T7RdhA*) exhibit relatively high consistency (Fig 7A). We also compared the distributions of the secondary structure elements in both *Nvjp-1* and *T7RdhA* in their respective 1000 AF2 models, as shown in S6 Fig in the SI. The *Nvjp-1* residues are mostly in the coil or turn states, whereas significant ratios of helices and sheets structures are observed in *TRdhA* residues.

Nevertheless, we found that the pLDDT profiles from AF2 models of both IDP (*Nvjp-1*) and ordered protein (*T7RdhA*) are highly consistent (Fig 7B). As discussed earlier that the pLDDT scores in AF2 models reflect the fluctuations of the residues [57], we suggest that, despite it is difficult to construct an ensemble of conformations to describe the characteristics

and functions of an IDP [58], the pLDDT profile may serve as a graphic, descriptive attribute, or feature.

In AF2 modeling, no multi-sequence alignment (MSA) hits was observed for *Nvjp-1*, i.e., it may be an orphan protein, or an orphan IDP. A recent language model RGN2 claimed that for the orphan proteins, it beats both AF2 [18] and RoseTTAFold [59] in both accuracy and speed [50]. The *Nvjp-1* models generated by RGN2 showed no apparent secondary structures (S7 Fig in the SI), in agreement with the AF2 models. Despite the potential "orphan" state and the intrinsic disorder of *Nvjp-1*, recent studies showed that, however, protein structures predicted from randomized protein sequences possess significant amount of ordered regions [57, 60]. This suggest that intrinsically disordered proteins do not appear by chance.

### 3.6. Ordered versus disordered, protein dynamics and AF2 models

As shown above, the residue-residue interactions in *Nvjp-1* are mainly transient. Even for different *Nvjp-1* AF2 models, we show that the majority of the calculated RINs are also transient. In contrast, the RINs in the AF2 models of the relatively well-folded protein, T7RdhA, are mainly persistent (Fig 8). For each RIN, the persistency is defined as the percentage of configurations in which the interaction can be observed. The *Nvjp-1* models has more residue-residue interactions than *T7RdhA* (Table 1), however, many of these interactions present in only a limited number of models (or a single model), but are absent in other models. Persistent interactions in *Nvjp-1* (Fig 8C, red bar) are mainly contributed by the peptide bonds between two

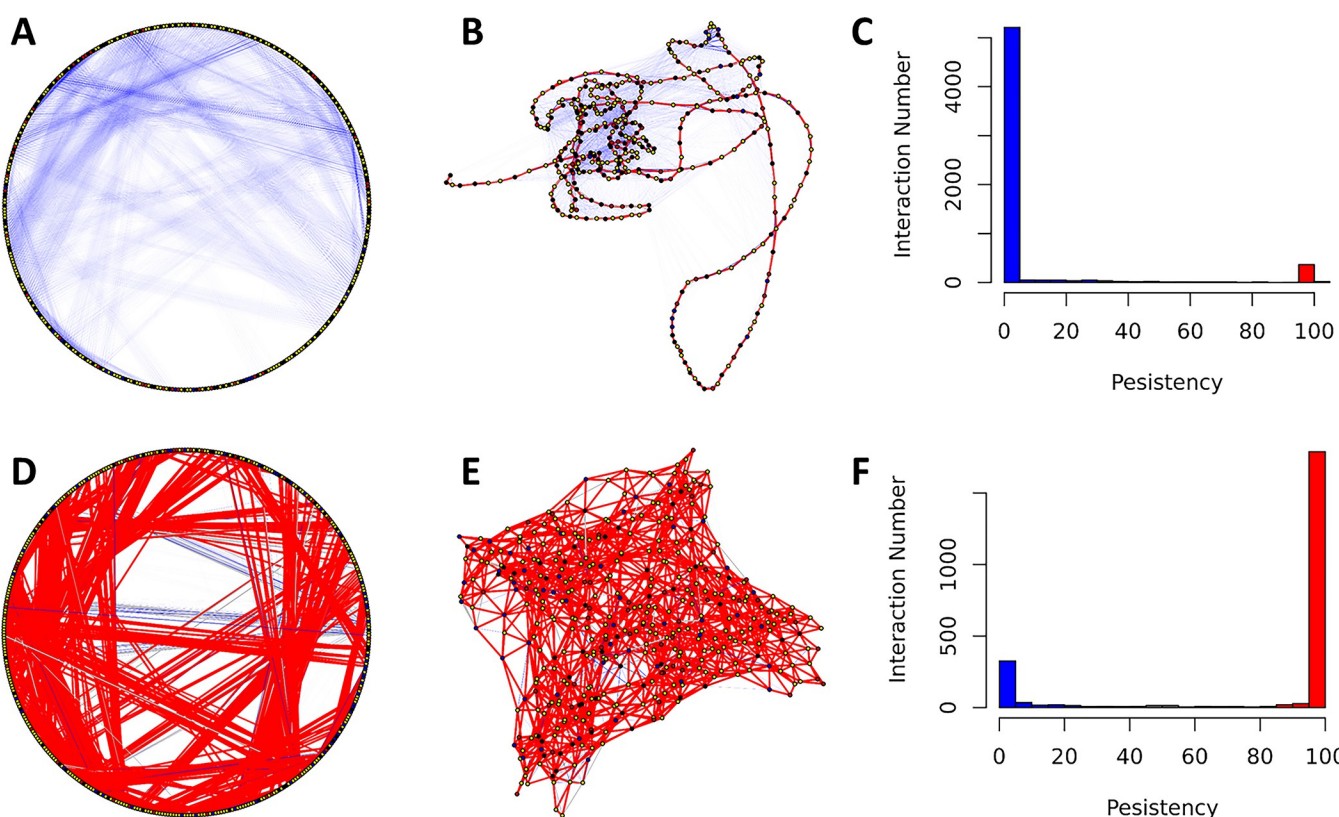

**Fig 8.** Ordered versus disordered: persistent and transient interactions in *Nvjp-1* (A-C, top) and *T7RdhA* (D-F, bottom). Circular (A and D) and randomized (B and E) RIN models are shown in which the persistent interactions (persistency ≥ 75%) are in red and transient interactions (persistency ≤ 25%) are in blue, and other interactions in gray. Histograms of the persistency (percentages) are also shown, similarly persistent in red and transient in blue.

consequential residues. Instead, in T7RdhA (Fig 8F), majority of the interactions are persistent, and only a small number of interactions are transient.

We further analyzed the persistency of residue-residue interactions throughout the MD simulations (also see Figs 3 and 6). In this analysis, we take snapshots every single ns during the last 500 ns MD trajectories of either the monomer or dimer simulation, and the RINs were calculated based on these 500 configurations. Noticeable differences between AF2 models (Fig 8A–8C) and MD configurations (Fig 9A–9C) can be found, i.e., there are apparently more persistent interactions from MD trajectory than those in the AF2 models. The main reason contributing to this differences might be that all AF2 models have been "relaxed" using energy minimizations and hence represent locally optimized (yet static) structures on the potential energy surface. For the configurations from MD, however, they are in constantly moving states governed by physical laws, and the persistency of RINs is restrained by the interaction potentials which may hold certain interactions (e.g., salt-bridges) longer than others. Nevertheless, the RINs in T7RdhA are highly persistent (Fig 8F), in line with the high pLDDT scores predicted by AF2 (Fig 7B). For *Nvjp-1*, the dominancy of transient RINs (Fig 8C) is also consistent with the low pLDDT scores (Fig 7B). Therefore, the pLDDT scores can be used to interpret the confidence of the structures, as well as residue-residue interactions.

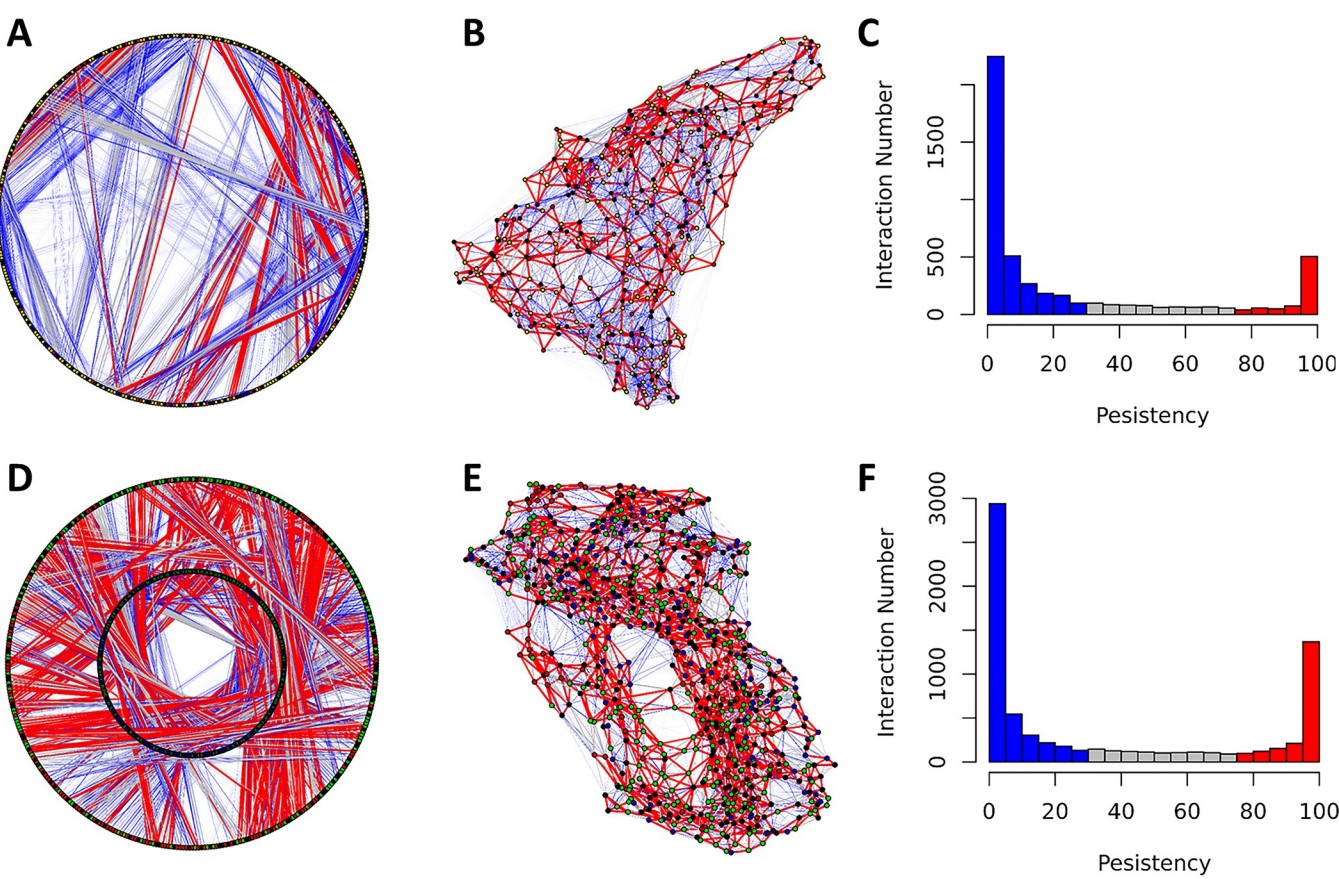

**Fig 9.** Residue-residue interaction networks from MD for Nvjp-1 monomer (A-C, HSE) and dimer (D-F, HSE).

## 4. Conclusions and future directions

Protein disorder is related to many factors such as flexibility [61], fuzziness [13], and entropy [60]. The residual disorder contents can be predicted from the protein sequence and hence are "intrinsic" [6]. Here, aided by AF2 and MD simulations, we propose an interesting aspect of protein disorder: 1) structure prediction is not reproducible by AF2, i.e., two independent AF2 predictions do not converge on a solution such as two models with low RMSD (at the atomic resolution); 2) the configurations from MD simulations do not overlap with each other (e.g., RMSD < 1 Å). The target of the present work, *Nvjp-1*, is an IDP in both monomeric and dimeric forms. Despite the relatively confined dynamics in the dimers, the *Nvjp-1* protein models (either from AF2 or from MD) are not similar to each other in both monomers and dimers, i.e., they are disordered. Using 1000 AF2 models, *Nvjp-1* is compared with *T7RdhA*, a well-folded globular protein. Unlike *Nvjp-1*, AF2 models of *T7RdhA* are consistent with the RMSD of ~0.5 Å. We observed that for both proteins, either disordered (*Nvjp-1*) or ordered (*T7RdhA*), AF2 produces highly consistent pLDDT profiles for all 1000 models, rendering it a potentially useful feature of the protein structures, or disorders. Moreover, in *Nvjp-1* AF2 models the residue-residue interactions are mostly transient, whereas the majority of those in *T7RdhA* are persistent.

The consistent residual pLDDT profiles predicted by AF2 for both disordered and ordered proteins suggest that the residual flexibilities and related protein dynamics may be encoded in the protein sequences [57]. Note that the 1000 AF2 models of *Nvjp-1* and *T7RdhA* were constructed by multiple AF2 runs with variable random seeds. It was suggested that this approach may uncover alternative functional configurations [18], including rare configurations [37]. With the recent advances, especially AF-cluster [62], we will be able to explore deeper in the folding spaces of interested proteins. Moreover, the physical adjacency-based RIN presented in this work will be applied to other systems, including the protein-ligand and protein-protein interactions, with potential applications to drug discoveries.

## Supporting information

**S1 Fig. The amino acid sequence of *Nvjp-1*.**
(TIF)

**S2 Fig. Circular representation of the residue interaction networks.**
(TIF)

**S3 Fig. Residue interaction networks from 100 ns MD trajectories of AF2 models.**
(TIF)

**S4 Fig. Comparison of the single chains in 25 *Nvjp-1* homodimers.**
(TIF)

**S5 Fig. Residue interaction network from the MD trajectory of *Nvjp-1* dimer models C.2 and D.0.**
(TIF)

**S6 Fig. Percentages of secondary structure elements (SSEs) estimated in 1000 AF2 models.**
(TIF)

**S7 Fig. The RGN2 model of *Nvjp-1* monomer.**
(TIF)

## Acknowledgments

The structural modeling and MD simulations were performed using the DoD HPC.

## Author Contributions

**Conceptualization:** Hao-Bo Guo, Rajiv Berry.

**Data curation:** Hao-Bo Guo, Baxter Huntington, Alexander Perminov, Kenya Smith, Nicholas Hastings, Rajiv Berry.

**Formal analysis:** Hao-Bo Guo.

**Investigation:** Rajiv Berry.

**Methodology:** Hao-Bo Guo.

**Project administration:** Nancy Kelley-Loughnane.

**Resources:** Nancy Kelley-Loughnane, Rajiv Berry.

**Software:** Hao-Bo Guo.

**Supervision:** Rajiv Berry.

**Validation:** Rajiv Berry.

**Visualization:** Hao-Bo Guo.

**Writing – original draft:** Hao-Bo Guo.

**Writing – review & editing:** Baxter Huntington, Alexander Perminov, Patrick Dennis, Nancy Kelley-Loughnane, Rajiv Berry.

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
