## [Decision Letter · Decision Letter 0]

12 Feb 2024

PONE-D-23-40853AlphaFold2 modeling and molecular dynamics simulations of an intrinsically disordered proteinPLOS ONE

Dear Dr. Guo,

Thank you for submitting your manuscript to PLOS ONE. After careful consideration, we feel that it has merit but does not fully meet PLOS ONE’s publication criteria as it currently stands. Therefore, we invite you to submit a revised version of the manuscript that addresses the points raised during the review process.

We look forward to receiving your revised manuscript.

Kind regards,

Arabinda Ghosh

Academic Editor

PLOS ONE

Journal Requirements:

"This work was supported by funding from the OUSD (R&E) ARAP Program. The structural 

modeling and MD simulations were performed using the DoD HPC."

"This work was supported by funding from the OUSD (R&E) ARAP Program."

5. Please expand the acronym “OUSD (R&E)” (as indicated in your financial disclosure) so that it states the name of your funders in full.

6. Thank you for stating the following financial disclosure: 

"This work was supported by funding from the OUSD (R&E) ARAP Program."

7. We note that you have referenced ([32,33] observed a potential dimer band from SDS-PAGE, which can sustain high temperature in SDS detergent (unpublished).) which has currently not yet been accepted for publication. Please remove this from your References and amend this to state in the body of your manuscript: (ie “Bewick et al. [Unpublished]”) as detailed online in our guide for authors

Reviewers' comments:

Reviewer's Responses to Questions

**Comments to the Author**

1. Is the manuscript technically sound, and do the data support the conclusions?

Reviewer #1: Yes

Reviewer #2: Yes

2. Has the statistical analysis been performed appropriately and rigorously? 

Reviewer #1: Yes

Reviewer #2: Yes

3. Have the authors made all data underlying the findings in their manuscript fully available?

Reviewer #1: Yes

Reviewer #2: Yes

4. Is the manuscript presented in an intelligible fashion and written in standard English?

Reviewer #1: Yes

Reviewer #2: Yes

5. Review Comments to the Author

Reviewer #1: The study utilizes various modeling and simulation techniques to explore the structural heterogeneity and disorder associated with Nvjp-1. The research delves into the unique characteristics of Nvjp-1, such as its modulated behavior by Zn-binding and its implications for dynamic sclerotization. Additionally, the study aims to understand the implications of disorder in the structural heterogeneity observed in the modeling and simulations of Nvjp-1.

- The manuscript is generally well-written and easy to follow. However, it would be beneficial to provide a clearer structure in the introduction section, outlining the main objectives and hypotheses of the study.

- Residue number should be mentioned for all the models in the main text

- Authors may calculate the one-dimensional (1D) distance variations (DV) between the Cɑ carbons to find the DV matrix from MD is highly consistent with the PAE matrix from AF2, indicating that the PAE matrix originates from protein dynamics

- Suggest potential future directions for research based on the current findings. This could include experimental validations, further computational studies, or applications in drug discovery.

Reviewer #2: Appreciable work has been carried out so far. However, the author can investigate the origin of Nvjp-1 from the phylogeny perspective in order to compare its closest protein sequence having an ordered form of arrangement that is meant for a specific function.

6. PLOS authors have the option to publish the peer review history of their article (what does this mean?). If published, this will include your full peer review and any attached files.

Reviewer #1: **Yes: **VINOD KUMAR YATA

Reviewer #2: No

---

## [Author Response · Author response to Decision Letter 0]

21 Feb 2024

Response: Checked. The PLoS style of references have been used for this revision.

Response: Checked. 

Response: Checked.

"This work was supported by funding from the OUSD (R&E) ARAP Program. The structural modeling and MD simulations were performed using the DoD HPC."

"This work was supported by funding from the OUSD (R&E) ARAP Program."

Response: We added the Funding Statement section in this revision. Thank you!

5. Please expand the acronym “OUSD (R&E)” (as indicated in your financial disclosure) so that it states the name of your funders in full.

Response: We expanded the acronym in this revision. Thank you!

6. Thank you for stating the following financial disclosure: 

"This work was supported by funding from the OUSD (R&E) ARAP Program."

Response: We added the suggested statement in this revision: 

“This work was supported by funding from the Office of the Under Secretary of Defense for Research and Engineering (OUSD R&E), Applied Research for Advancement of S&T Priorities (ARAP) Program. The funders had no role in study design, data collection and analysis, decision to publish, or preparation of the manuscript.”

Thank you!

7. We note that you have referenced ([32,33] observed a potential dimer band from SDS-PAGE, which can sustain high temperature in SDS detergent (unpublished).) which has currently not yet been accepted for publication. Please remove this from your References and amend this to state in the body of your manuscript: (ie “Bewick et al. [Unpublished]”) as detailed online in our guide for authors

Response: We revised as your suggestion and cited the SDS-PAGE work as (Dennis et al., unpublished). 

Thank you for this suggestion!

Response: We listed the “Supporting Information Appendix” at the end of the MS, before the references. Thank you!

Response: Checked, thank you!

Reviewers' comments:

Reviewer's Responses to Questions

Comments to the Author

1. Is the manuscript technically sound, and do the data support the conclusions?

Reviewer #1: Yes

Reviewer #2: Yes

2. Has the statistical analysis been performed appropriately and rigorously?

Reviewer #1: Yes

Reviewer #2: Yes

3. Have the authors made all data underlying the findings in their manuscript fully available?

Reviewer #1: Yes

Reviewer #2: Yes

4. Is the manuscript presented in an intelligible fashion and written in standard English?

Reviewer #1: Yes

Reviewer #2: Yes

5. Review Comments to the Author

Reviewer #1: The study utilizes various modeling and simulation techniques to explore the structural heterogeneity and disorder associated with Nvjp-1. The research delves into the unique characteristics of Nvjp-1, such as its modulated behavior by Zn-binding and its implications for dynamic sclerotization. Additionally, the study aims to understand the implications of disorder in the structural heterogeneity observed in the modeling and simulations of Nvjp-1.

- The manuscript is generally well-written and easy to follow. However, it would be beneficial to provide a clearer structure in the introduction section, outlining the main objectives and hypotheses of the study.

Response: Thank you for this suggestion! 

We have proposed the main objectives in the Introduction section, i.e., what defines an IDP. These include the sequence-based ID predictions, the structural heterogeneity from multiple AF2 models, and the distinct patterns in the residue-residue interaction network compared to other well-folded proteins. In the revision, we added a description paragraph at the end of the Introduction section to outline the structure of this manuscript.

- Residue number should be mentioned for all the models in the main text

Response: Thank you! 

This work mainly discuss the intrinsically disordered protein, Nvjp-1, which has 381 AA, and a well-ordered protein, T7RdhA, which has 406 AA. The protein sequence of Nvjp-1 have been listed in the supporting information (Fig. S1). The residue number information can also be found in the Figure 7 caption. In this revision, we do added the amino acid numbers of the models in the main text.

- Authors may calculate the one-dimensional (1D) distance variations (DV) between the Cɑ carbons to find the DV matrix from MD is highly consistent with the PAE matrix from AF2, indicating that the PAE matrix originates from protein dynamics

Response: Thank you for this suggestion!

In many previous related modeling, including AF2 and RoseTTAFold, the distance matrices are usually measured using the distances between the C� carbons (C� for Gly), as the main focuses are usually the functions and/or dynamics from the protein sidechains. We used this matrix in our previous analysis. However, we found this approach will lead to apparent errors. Please find the examples in Fig. S6 of the supplementary information in this paper: https://www.nature.com/articles/s41598-023-30310-x.

It is the main reason we think the approach used in the present work is reliable.

- Suggest potential future directions for research based on the current findings. This could include experimental validations, further computational studies, or applications in drug discovery.

Response: Thank you so much!

This is really a great suggestion! We changed Section 4 to “Conclusions and Future Directions”. We suggest that the recent advances, especially AF-cluster, can be applied to explore the protein folding spaces that may result in alternative insights to the protein functions; we also plan to use our approach in other directions including protein-ligand and protein-protein interactions, with the potential applications to drug discovery.

Reviewer #2: Appreciable work has been carried out so far. However, the author can investigate the origin of Nvjp-1 from the phylogeny perspective in order to compare its closest protein sequence having an ordered form of arrangement that is meant for a specific function.

Response: We thank you for this suggestion! We did mention in this paper that Nvjp-1 is an “orphan” protein with no closely related BLAST hits available. Therefore, we are not able to provide the phylogenetic insights of this protein so far. A deeper investigation on the phylogeny of Nvjp-1 or other IDPs, is out of the scoop of the present work. But we believe in the future, with the continuous development in both the protein sequences and structures, we might have the opportunity to complete this analysis to explore the functions and applications of the Nvjp-1 and Nvjp-1 related proteins.

---

## [Decision Letter · Decision Letter 1]

25 Mar 2024

AlphaFold2 modeling and molecular dynamics simulations of an intrinsically disordered protein

PONE-D-23-40853R1

Dear Dr. Hao-Bo Guo,

We’re pleased to inform you that your manuscript has been judged scientifically suitable for publication and will be formally accepted for publication once it meets all outstanding technical requirements.

Kind regards,

Arabinda Ghosh

Academic Editor

PLOS ONE

Additional Editor Comments (optional):

Reviewers' comments:

Reviewer's Responses to Questions

**Comments to the Author**

1. If the authors have adequately addressed your comments raised in a previous round of review and you feel that this manuscript is now acceptable for publication, you may indicate that here to bypass the “Comments to the Author” section, enter your conflict of interest statement in the “Confidential to Editor” section, and submit your "Accept" recommendation.

Reviewer #1: All comments have been addressed

2. Is the manuscript technically sound, and do the data support the conclusions?

Reviewer #1: Yes

3. Has the statistical analysis been performed appropriately and rigorously? 

Reviewer #1: Yes

4. Have the authors made all data underlying the findings in their manuscript fully available?

Reviewer #1: Yes

5. Is the manuscript presented in an intelligible fashion and written in standard English?

Reviewer #1: Yes

6. Review Comments to the Author

Reviewer #1: (No Response)

---

## [Editor Report · Acceptance letter]

26 Apr 2024

PONE-D-23-40853R1 

PLOS ONE

Dear Dr. Guo, 

I'm pleased to inform you that your manuscript has been deemed suitable for publication in PLOS ONE. Congratulations! Your manuscript is now being handed over to our production team.

Kind regards, 

on behalf of

Dr. Arabinda Ghosh 

Academic Editor

PLOS ONE